# Incommensurate smectic phase in close proximity to the high-$T_c$ superconductor FeSe/SrTiO$_3$

Yonghao Yuan[1,2], Xuemin Fan[1,2], Xintong Wang[1,2], Ke He [1,2,3], Yan Zhang [4,5], Qi-Kun Xue[1,2,3✉] & Wei Li [1,2✉]

Superconductivity is significantly enhanced in monolayer FeSe grown on SrTiO$_3$, but not for multilayer films, in which large strength of nematicity develops. However, the link between the high-transition temperature superconductivity in monolayer and the correlation related nematicity in multilayer FeSe films is not well understood. Here, we use low-temperature scanning tunneling microscopy to study few-layer FeSe thin films grown by molecular beam epitaxy. We observe an incommensurate long-range smectic phase, which solely appears in bilayer FeSe films. The smectic order still locally exists and gradually fades away with increasing film thickness, while it suddenly vanishes in monolayer FeSe, indicative of an abrupt smectic phase transition. Surface alkali-metal doping can suppress the smectic phase and induce high-$T_c$ superconductivity in bilayer FeSe. Our observations provide evidence that the monolayer FeSe is in close proximity to the smectic phase, and its superconductivity is likely enhanced by this electronic instability as well.

[1] State Key Laboratory of Low-Dimensional Quantum Physics, Department of Physics, Tsinghua University, Beijing, China. [2] Frontier Science Center for Quantum Information, Beijing, China. [3] Beijing Academy of Quantum Information Sciences, Beijing, China. [4] International Centre for Quantum Materials, School of Physics, Peking University, Beijing, China. [5] Collaborative Innovation Centre of Quantum Matter, Beijing, China. ✉email: qkxue@mail.tsinghua.edu.cn; weili83@tsinghua.edu.cn

E lectronic liquid crystal phases are correlation-induced intermediate states between the weak coupling Fermi liquid and localized electronic crystals[1,2]. The symmetries of electronic structures in these states are lower than those of the lattice. For example, the nematic phase spontaneously breaks rotational symmetry and the smectic phase further reduces translational symmetry. The electronic liquid crystal phases have attracted broad interest since they widely appear and seem to be intrinsic in cuprates[3–14], iron-based superconductors[15–24], and topological quantum meterials[25]. In iron-based superconductors, the electronic anisotropy is too large to be understood if one solely considers the lattice degree of freedom. Instead, the charge, spin, and orbital degrees of freedom are essential for the development of nematicity[16,17,19,26–28]. Recent studies show that the optimal doping levels of the high-transition temperature (high-$T_c$) superconductors usually correspond to the nematic quantum critical points (QCP), at which the nematic fluctuations are optimized[13,29]. These electronic instabilities are believed to play some key roles in realization of high-$T_c$ superconductivity[30,31].

FeSe is a good material to investigate the electronic liquid crystal phases because it has the simplest structure among iron-based superconductors and, unlike others, shows large separation between nematic and long-range antiferromagnetic (AFM) states in phase diagram[32–36]. High-pressure measurements reveal a two-step superconductivity enhancement at nematic and magnetic QCP, and $T_c$ is optimized to 38 K from its bulk value 8.5 K[32,35,36]. More intriguingly, one unit-cell (UC) FeSe thin film grown on SrTiO$_3$ (FeSe/STO) shows a distinct new high-$T_c$ superconducting phase with the $T_c$ up to 65 K[37–41]. The underneath STO substrate contributes crucially to the modification of electronic structure and the boost of $T_c$. It provides not only tensile strain and carrier transfer, but also additional electron–phonon coupling channel[42] to FeSe films. Surprisingly, such a high-$T_c$ superconducting phase only exists in 1 UC FeSe and is missing for thicker films up to 100 nm[43]. Meanwhile, the strength of nematicity is enlarged in those non-superconducting multilayer FeSe films with tensile strain[39]. The nematic transition temperature increases from 120 to 170 K when the thickness of the film decreases from 30 to 2 UC[21,24,39], much higher than its bulk value 90 K. More importantly, in these multilayer FeSe/STO, besides the enhanced strength of nematicity, short-range stripes, which further breaks translational symmetry, are observed in the vicinity of defects[21], indicating rather strong smectic fluctuation as well as electronic correlation. Compared with bulk FeSe, the enhancement of electronic correlation in multilayer FeSe/STO is expected due to its low-dimensionality and lattice expansion[1]. However, a direct connection between the correlation enhanced nematic state in multilayer films and the high-$T_c$ superconducting state in monolayer film has not yet been revealed.

Scanning tunneling microscopy (STM) is a powerful tool to investigate the electronic structure of quantum materials in real space. It can capture the surface topography in atomic scale, and measure the local density of states (DOS) of quasiparticles by scanning tunneling spectroscopy (STS or d$I$/d$V$ spectra), and has been used to study the electronic liquid crystal phases in high-$T_c$ systems[5,6,8,10,12,14,18,19,21,23].

In this letter, we present our low-temperature STM study on few-layer FeSe films grown on STO. We find a unique smectic electronic phase in 2 UC FeSe/STO, which manifests as incommensurate long-range stripe pattern as seen in STM d$I$/d$V$ mapping. This stripe pattern, reminiscent of the checkerboard structure in cuprate superconductors, is rarely found in iron-based superconductors, except for uniaxial strained LiFeAs[22]. The long-range ordering degenerates to short-range smectic fluctuations in 3 UC and thicker FeSe films, but vanishes in 1 UC FeSe, indicating the occurrence of an abrupt phase transition. Such

smectic instability may provide additional superconductivity enhancement in 1 UC FeSe films.

## Results

**Incommensurate smectic phase in 2 UC FeSe/STO.** Figure 1b exhibits a typical d$I$/d$V$ mapping taken on a 2 UC FeSe film, in which long-range stripe patterns are clearly observed. Compared with the nematic phase of multilayer FeSe films, in which short-range stripes are observed only in the vicinity of defects[21], the long-range stripe pattern observed here globally breaks the rotational and translational symmetries of the lattice, therefore corresponding to a well-developed smectic electronic phase.

Smectic domains are formed on the surface. The stripe patterns in adjacent domains are perpendicular to each other (see the double-headed white arrows in Fig. 1b). The domains are separated by smectic domain walls, which manifest themselves as white wrinkles with higher differential conductance (Fig. 1b). Within the domain regions, another kind of DOS corrugation appears, denoted by the white dashed lines in Fig. 1b, c. Compared with the smectic domain walls, they show less contrast and have no effect on the orientations of the stripe patterns. This corrugation actually originates from a kind of structure-related boundaries in the underlying 1 UC FeSe film, which separates 2 × 1 reconstruction domains[44] (also see Supplementary Note 1). The influence of those boundaries on DOS penetrates into the above FeSe layer and is captured by STM. Interestingly, although the smectic phase and the 2 × 1 reconstruction both break rotational symmetry, the routes of these two kinds of boundaries are irrelevant on the 2 UC FeSe surface, implying that the structural transition is not the driving force for the smectic phase.

Figure 1c shows an atomically resolved STM topographic image of a 2 UC FeSe film. The surface is Se-terminated (see the schematic in Fig. 1a) and the stripe pattern is along the diagonal direction of Se–Se lattice, i.e., the underlying Fe–Fe lattice direction. The stripes have a spatial period of $\lambda = 2.0$ nm, corresponding to a wave vector $q_0 = 0.19 q_{Se}$ in the fast Fourier transformation (FFT) image (inset of Fig. 1c), where $q_{Se}$ is the wave vector of the Bragg point of Se–Se lattice. The period of the stripes is incommensurate, and it shows slight fluctuations in different regions (1.9–2.1 nm). This is consistent with the theoretical proposal of smectic ordering at finite temperatures[1]. The orientation and period of the long-range stripes observed here are the same as those of the short-range stripes pinned by defects in the multilayer FeSe films[21], demonstrating that the smectic fluctuation is stabilized in 2 UC FeSe film. These long-range stripes are able to exist in defect-free regions, indicating the development of the smectic phase[1,2].

The energy dependence of the stripe pattern is also investigated. The main panel of Fig. 1d presents the d$^3I$/d$V^3$ values as a function of energy and position or named as $v(x, E)$. The route (the line-cut), along which the d$^3I$/d$V^3$ values are extracted, is marked by the red dashed arrow in Fig. 1b. Second-order derivative is performed to the raw d$I$/d$V$ data to diminish the absolute differential conductance amplitude at different energies so that the energy-dependent stripes are clearly drawn in the upper panel of Fig. 1d, in which the information of the period and phase of the stripes can be easily obtained (the raw data are shown in Supplementary Fig. 2). From the $v(x, E)$ image, we find the stripe ordering is most pronounced in three energy ranges, which are −100 to −30 meV, 30 to 60 meV, and 90 to 100 meV (within the blue, green and red dashed rectangles, respectively). The averaged d$^3I$/d$V^3$ line-cut in those energy ranges are plotted in the right panel of Fig. 1d, in which the stripes in 30–60 meV show a π phase shift compared with those in other two energy ranges. The periods of the stripes at different

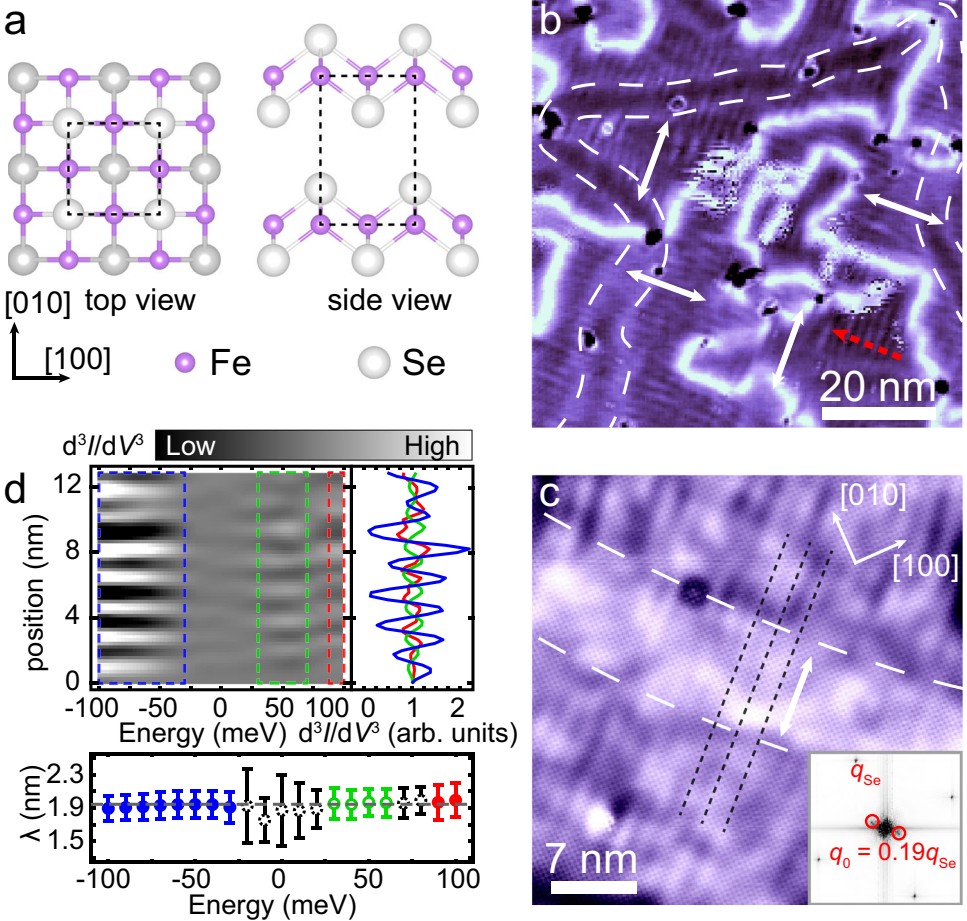

**Fig. 1 Stripe patterns in 2 UC FeSe/STO. a** Lattice structure of FeSe. **b** A STM $dI/dV$ mapping of a 2 UC FeSe/STO at 150 meV (76 nm × 76 nm; set point, $V_s = 150$ mV, $I_t = 400$ pA). The double-headed white arrows denote the orientation of stripe ordering in different smectic domains. **c** Atomically resolved topographic image of the 2 UC FeSe/STO (35 nm × 35 nm; set point, $V_s = 60$ mV, $I_t = 200$ pA). The black dashed lines denote three adjacent stripes. The white dashed lines denote the DOS corrugations induced by structural boundaries in the underneath FeSe layer. The stripe patterns are continuous across the corrugation. Inset: Fast Fourier transformation result of (**c**). The $q_O = 0.19q_{Se}$ is observed along the [−110] direction, corresponding to the stripes with a period of $\lambda = 2.0$ nm in real space. **d** Upper: The second-order derivative of $dI/dV$ (i.e., $d^3I/dV^3$) line-cut as a function of energy [or $v(x, E)$]. The corresponding route of the line-cut is shown in red dashed arrow in (**b**). Right panel: The blue, green, and red curves are the averaged $d^3I/dV^3$ line-cuts in the energy ranges of −100 to −30 meV, 30 to 60 meV, and 90 to 100 meV, respectively. The stripes show a $\pi$ phase shift in the range of 30–60 meV. Lower panel: The period of the stripes as a function of energy, which is determined from the peak position in the FFT result to the $d^3I/dV^3$ line-cut at each energy. The error-bar denotes the peak width in the FFT. The hollow green circles denote the stripes have a $\pi$ phase shift compared with those labeled in filled circles. The data labeled in gray dashed circles are in the transition energy ranges, in which the signals are relatively weak. And the phase shift in such ranges cannot be determined.

energies are identical, as shown in the lower panel of Fig. 1d. The non-dispersive behavior confirms that the stripe patterns originate from a static smectic electronic modulation rather than quasiparticle interference.

**Smectic phase at different thickness**. Previous study demonstrates that the long-range stripes are absent in 30 UC FeSe[21]. Hence, between 2 UC and 30 UC, there must be an electronic structure transition. To study the transition, we perform thickness-dependent measurements. We focus on a specific area of a FeSe film, in which both 1 UC and 2 UC FeSe are included (upper panel of Fig. 2a). $dI/dV$ spectra taken on them (lower panel of Fig. 2a) clearly show the superconducting and non-superconducting behaviors of 1 UC and 2 UC FeSe, respectively. To reveal the stripes, $dI/dV$ mappings Fig. 2b–d are taken at different energies on the same area of Fig. 2a. As shown in Fig. 2b–d, the stripes as well as domain walls suddenly disappear at the step edge between 2 UC and 1 UC FeSe (highlighted by

orange dashed lines) and they do not survive in 1 UC FeSe any more, indicating a complete suppression of the electronic liquid crystal phase. In principle, the further enhanced lattice expansion in 1 UC FeSe (due to its close proximity to STO) should lead to larger strength of nematicity as well as the stripes[39]. However, the extra itinerant electrons, provided by STO substrate, may give rise to the suppression of the electronic liquid crystal phase in 1 UC FeSe.

Intriguingly, the long-range stripe pattern disappears in 3 UC FeSe as well. Figure 2e shows an area consisting of 2 UC and 3 UC FeSe. In the corresponding $dI/dV$ mappings (Fig. 2f–h), the domain walls cross the step edge and continuously propagate on the 2 UC and 3 UC FeSe. The domain walls on 3 UC and 2 UC FeSe are the boundaries of nematic and smectic domains, respectively. The continuous propagation of the domain walls here indicates the close relationship between the nematic phase and the smectic phase in FeSe/STO. The nematicity (as well as nematic domain walls) appears first. Then smectic states may develop based on nematicity (under certain conditions), and thus

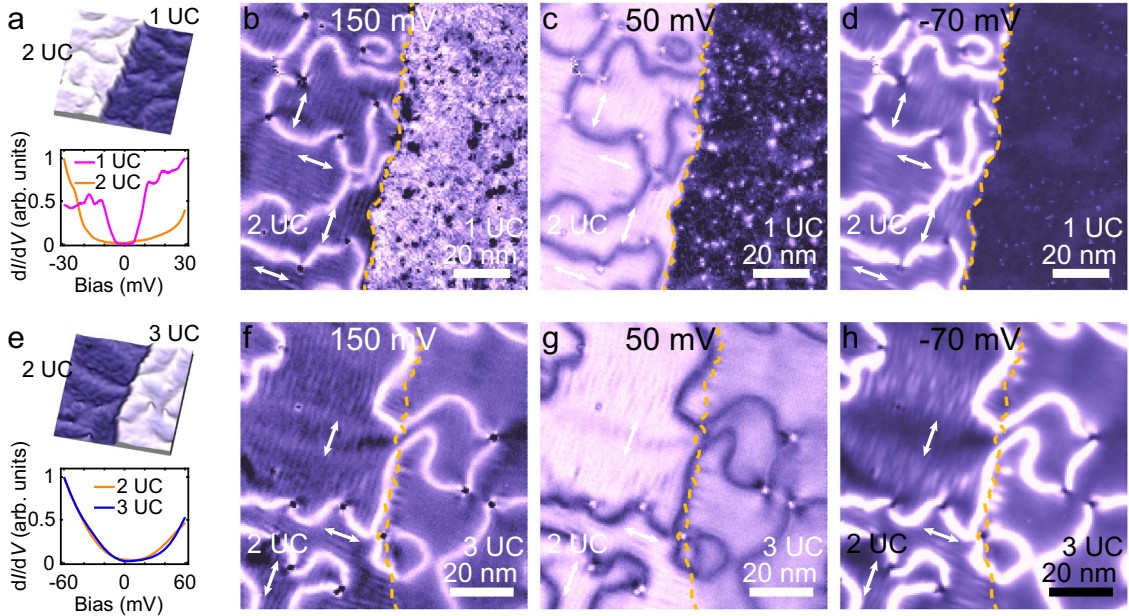

**Fig. 2 Film thickness dependence of smectic phase. a** Upper panel: STM topographic image of an area including 1 UC and 2 UC FeSe/STO (90 nm × 90 nm; set point, $V_s = 60$ mV, $I_t = 200$ pA). Lower panel: d$I$/d$V$ spectra of 1 UC and 2 UC FeSe (set point, $V_s = 30$ mV, $I_t = 300$ pA). The spectrum of 1 UC FeSe shows clear superconducting gap. **b**–**d** d$I$/d$V$ mappings (90 nm × 90 nm; set point, $V_s = 150$ mV, $I_t = 300$ pA) taken on the same area in (**a**) at bias voltages of 150, 50, and −70 mV, respectively. The orange dashed line denotes the step edge. The stripe patterns as well as the domain walls are absent in 1 UC FeSe. **e** Upper panel: STM topographic image of an area including 2 UC and 3 UC FeSe/STO (80 nm × 80 nm; set point, $V_s = 60$ mV, $I_t = 300$ pA). Lower panel: d$I$/d$V$ spectra of 2 UC and 3 UC FeSe (set point, $V_s = 500$ mV, $I_t = 200$ pA). Both 2 UC and 3 UC FeSe are not superconducting. **f**–**h** d$I$/d$V$ mappings (80 nm × 80 nm; set point, $V_s = 150$ mV, $I_t = 300$ pA) taken on the same area in (**e**) at bias voltages of 150, 50, and −70 mV, respectively. The domain walls are continuous on the step edge but the stripe patterns disappear on the 3 UC FeSe, indicating a smectic-to-nematic quantum phase transition from 2 UC to 3 UC FeSe.

inherit its domain walls. In contrast, the stripe patterns disappear at the step edge and are invisible in the 3 UC FeSe. These findings demonstrate that the long-range smectic phase is suppressed, while the nematic phase still persists in 3 UC FeSe. The nematicity-induced splitting of $d_{xz}$ and $d_{yz}$ bands in 3 UC and 2 UC FeSe has been detected by ARPES[24,39] and quasiparticle interference measurements[21,45] (see Supplementary Notes 3–5 and Movies 1–4). Similar to the case of 30 UC FeSe, the short-range stripes pinned by defects, indicative of smectic fluctuations, are visible in 3 UC FeSe (see Supplementary Fig. 8). Our observations indicate that FeSe thin films actually undergoes a smectic-to-nematic phase transition from 2 UC to 3 UC. The smectic phase in 2 UC FeSe melts at elevated temperature and degenerates to nematic phase (Supplementary Fig. 9).

**Doping dependence of smectic phase and superconductivity.** As mentioned, electron doping from the STO substrate is one of the key parameters that controls the electronic liquid crystal phases and superconductivity in different layers of FeSe. To investigate the doping effect, we deposit Rb atoms on the surface of a 2.5 UC FeSe sample (Supplementary Note 8), in which 2 UC and 3 UC films are coexisted. Figure 3 exhibits the topographic images of 2 UC FeSe with various Rb coverage from 0 to 0.0265 ML. Here, 1 monolayer (ML) is defined as 1 Rb atom per Fe site (approximately corresponds to 1 electron/Fe). The Rb atoms can locally suppress the stripe orderings. To be specific, as shown in Fig. 3b, the long-range stripe orderings apparently detour around the Rb atoms, which leads to a dramatic phase decoherence of the stripes in the regions with high Rb concentration. In the low Rb concentration regions, the stripes are still of long-range coherence. When the overall doping concentration increases (Fig. 3c, d), the stripe area gradually reduces. When the Rb coverage

increases to 0.0265 ML, the stripe ordering is totally suppressed (Fig. 3e). Figure 3f summarizes the suppression of stripes with increased Rb coverage, in which the ratios of stripe area are estimated from the topographic images (Fig. 3a–e, also see Supplementary Note 9). These results further support our viewpoint: the absence of smectic phase in 1 UC FeSe is due to the charge transfer from substrate.

Actually, superconductivity in 2 UC FeSe starts to emerge at the Rb coverage of 0.0265 ML. We also carry out systematic study of the Rb doping effect on superconductivity in 2 UC and 3 UC FeSe films (Supplementary Note 10).

Figure 4 summarizes the doping dependence of the superconducting gaps in 2 UC and 3 UC FeSe thin films. In our experiment, we randomly take 100 d$I$/d$V$ spectra at each doping level of 2 UC and 3 UC FeSe. We find that the superconducting gap presents inhomogeneity and the strength of inhomogeneity is related to the doping concentrations. Therefore, the superconductivity in Rb-coated FeSe films is characterized by two aspects: the homogeneity and the gap size.

The d$I$/d$V$ spectra taken at each doping level are sorted into two groups on the basis of their superconducting gaps (see Supplementary Note 11). The good superconducting group contains the spectra that show clean and symmetric (with $E_F$) superconducting gaps. The averaged spectra for good superconducting groups in 2 UC and 3 UC FeSe are shown in Fig. 4a and Fig. 4c, respectively. The bad superconducting group includes the spectra that show asymmetric superconducting gaps or contain in-gap states. The averaged spectra for bad superconducting groups in 2 UC and 3 UC FeSe are shown in Fig. 4b and Fig. 4d, respectively. In 2 UC FeSe, superconductivity emerges with 0.0265 ML Rb doping (Fig. 4a). While it does not appear in 3 UC FeSe until the Rb coverage increases to 0.038 ML (Fig. 4c).

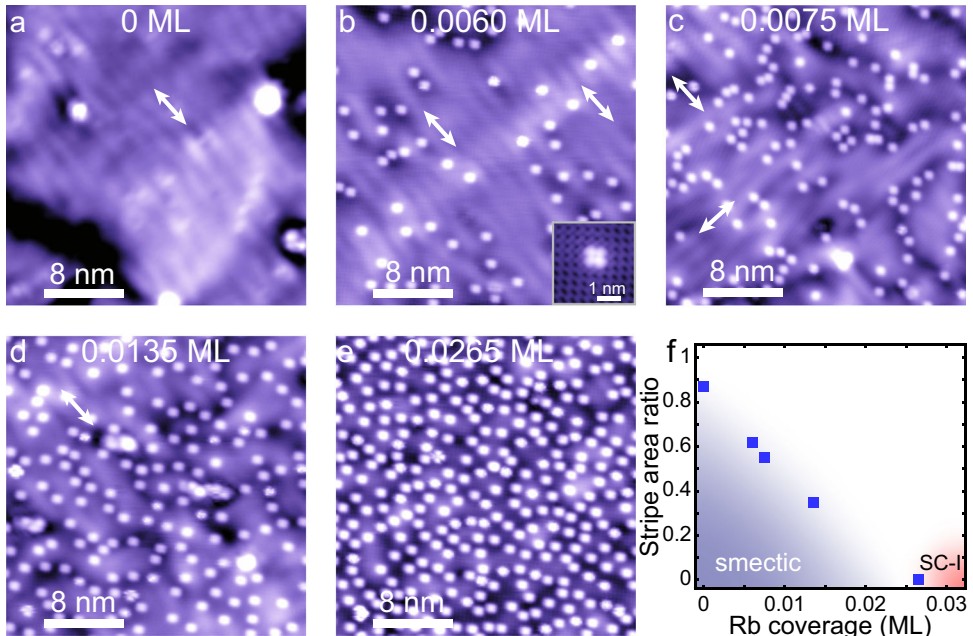

**Fig. 3 Suppression of smectic phase in 2 UC FeSe by Rb surface doping. a–e** Topographic images of 2 UC FeSe with surface Rb doping at various coverages: **a** 0 ML (30 nm × 30 nm; set point, $V_s = 60$ mV, $I_t = 30$ pA), **b** 0.0060 ML (30 nm × 30 nm; set point, $V_s = 60$ mV, $I_t = 50$ pA), **c** 0.0075 ML (30 nm × 30 nm; set point, $V_s = 60$ mV, $I_t = 40$ pA), **d** 0.0135 ML (30 nm × 30 nm; set point, $V_s = 100$ mV, $I_t = 20$ pA), **e** 0.0265 ML (30 nm × 30 nm; set point, $V_s = 60$ mV, $I_t = 50$ pA). Inset of **b**: STM topographic image of a single Rb atom adsorbed on 2 UC FeSe (see Supplementary Fig. 10). **f** Doping dependence of stripe area ratio (shown in blue dots) estimated from (**a**) to (**e**) (see Supplementary Fig. 11). The blue and red shaded regions denote the smectic and superconducting states, respectively.

Here, we define a good-superconductivity ratio as the proportion of the $dI/dV$ spectra showing good superconducting gap at each Rb coverage. The good-superconductivity ratios are counted and shown in Figs. 4a and c at each doping level. The Rb coverage-dependent ratio curves of 2 UC and 3 UC FeSe are summarized in Fig. 4e. In both 2 UC and 3 UC FeSe thin films, the ratio increases with Rb coverage at the beginning and finally drops in the over-doped regime, exhibiting a dome-like feature. Comparing with 3 UC FeSe, 2 UC FeSe presents a higher maximum value of the homogeneity ratio (94.1% in 2 UC vs. 90.9% in 3 UC, also see the gray dashed guide line in Fig. 4e). In addition, the ratio curve of 2 UC FeSe shows a terrace-like shape, indicating that it has a wider doping range with high homogeneity of superconductivity. The maximum values of the ratios for 2 UC and 3 UC FeSe correspond to Rb coverage of 0.095 ML and 0.145 ML, respectively, at which the two averaged $dI/dV$ spectra show decent superconducting gap features (Fig. 4f). Compared with that of 3 UC FeSe, the averaged $dI/dV$ spectrum of 2 UC FeSe possesses much sharper coherence peaks (Fig. 4f), indicating even better superconductivity.

On the other hand, the superconducting gap sizes of each $dI/dV$ spectra of the good superconducting groups are extracted and taken into statistics. The doping dependence of gap size is shown in Fig. 4g. Superconductivity in 2 UC and 3 UC FeSe both present dome-like features. The gap size in optimally doped 2 UC FeSe (12.75 meV) is larger than that in 3 UC FeSe (11.56 meV), which again shows that superconductivity in 2 UC FeSe is better. Our finding presents similar result to that of K doped FeSe thin films[46,47].

By comparing the two key aspects of superconductivity that we mentioned before, the homogeneity and the gap size, we demonstrate that the dopant-induced superconductivity in 2 UC FeSe is even better than that in 3 UC FeSe. Given the fact that the main difference between 2 UC and 3 UC FeSe is whether the long-range smectic phase exists, the relationship between the

superconductivity and the semetic phase in FeSe is clearly revealed. Although the long-range stripes compete with the superconductivity in undoped 2 UC FeSe, as the stripes are suppressed by electron doping, the residual smectic fluctuations provide additional enhancement of superconductivity.

## Discussion

The electronic structures in different layers of FeSe films are schematically shown in Fig. 5a. The smectic phase, manifesting as long-range stripes, only exists in 2 UC FeSe. It is abruptly suppressed in the layers beneath and above 2 UC, in which the high-$T_c$ superconductivity and strong nematicity develops, respectively. Lattice expansion and charge transfer, which both originate from the STO substrate, are the two key parameters to control the electronic liquid crystal phases in FeSe thin films. Their influences spread to different length scales along the $z$-axis and generate various electronic structures. On one hand, the effect of charge transfer decays rapidly and only shows significant influence in 1 UC FeSe, giving rise to the enhancement of superconductivity[39]. On the other hand, the lattice expansion, despite decreasing in thicker FeSe films, is still observable even in the films as thick as 40 UC[39], leading to large electronic correlation and strong nematicity in multilayer FeSe/STO[21].

The unique smectic phase in 2 UC FeSe may arise from a delicate balance between lattice and charge degrees of freedom. First, the tensile strain in 2 UC FeSe is larger than that in 3 UC FeSe, giving rise to stronger nematicity and electronic correlation. Given the fact that short-range stripe patterns are observed in the vicinity of the defects in multilayer FeSe/STO, where larger strength of electronic anisotropy (nematicity) is also expected, the smectic transition thus could be promoted in 2 UC FeSe. Meanwhile, based on the ARPES data[39], a small amount of charge transfer still exists in 2 UC FeSe but is barely found in 3 UC FeSe. Therefore, the change of charge transfer between 2 UC and 3 UC FeSe is more dramatic than that of the tensile strain, and the

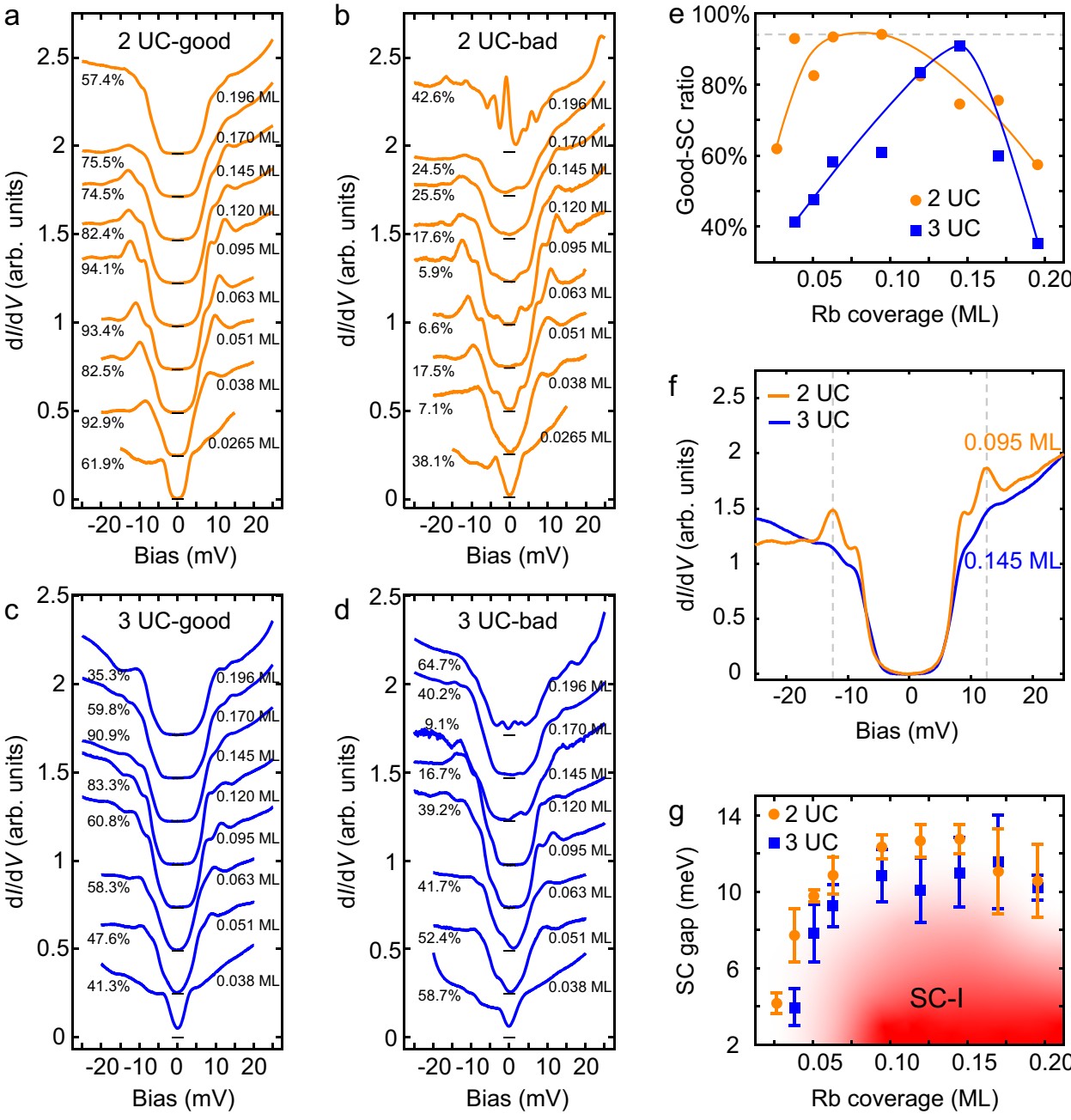

**Fig. 4 Superconductivity induced by Rb surface doping in 2 UC and 3 UC FeSe thin films. a**, **b** Doping dependence of averaged d$I$/d$V$ spectra taken on 2 UC FeSe with good and bad superconducting gaps, respectively. **c**, **d** Doping dependence of averaged d$I$/d$V$ spectra taken on 3 UC FeSe with good and bad superconducting gaps, respectively. Here, symmetric coherent peaks and absence of in-gap state are the criteria for good superconducting gap. In opposite, the spectra which show asymmetric superconducting gap or in-gap states are sorted into the bad superconducting group. **e** The dependence of good-superconductivity ratio on Rb coverage. Here, the ratio is defined as the proportion of the d$I$/d$V$ spectra showing good superconducting gap at each Rb coverage. The higher ratio indicates better homogeneity. The curves are guides to the eye. **f** Averaged d$I$/d$V$ spectra extracted from (**a**) and (**c**) at the doping levels with the optimal homogeneity in 2 UC and 3 UC FeSe. **g** Dependence of the superconducting gap size on Rb coverage. The spectra in good superconducting group are taken into statistics. The error bars denote the standard deviations of superconducting gap sizes at different Rb coverages.

unique charge transfer in 2 UC FeSe may play an important role to stabilize the smectic stripe patterns as well. It is worthy to note again that the tensile strain in 1 UC FeSe is even larger than that in 2 UC FeSe, which is propitious to form the stripe patterns, but the heavy electron doping from STO drives 1 UC FeSe into the high-$T_c$ superconducting state, which has also been simulated by the Rb surface doping on 2 UC FeSe. We therefore expect that smectic phase can also emerge once the dopants are appropriately removed from 1 UC FeSe/STO.

A promising way to decouple the charge and lattice degrees of freedom is to grow FeSe film on a substrate with an even larger lattice constant. BaTiO$_3$ (BTO) could be a good candidate. Since the strength of tensile strain can be gradually tuned by the thickness of FeSe film, systematic study of FeSe/BTO may provide sufficient information to understand the role of strains for formation of smectic phase. We are still working on the growth of FeSe films on BTO, however, it is challenging due to the large amount of defects on BTO substrate, which could be significantly

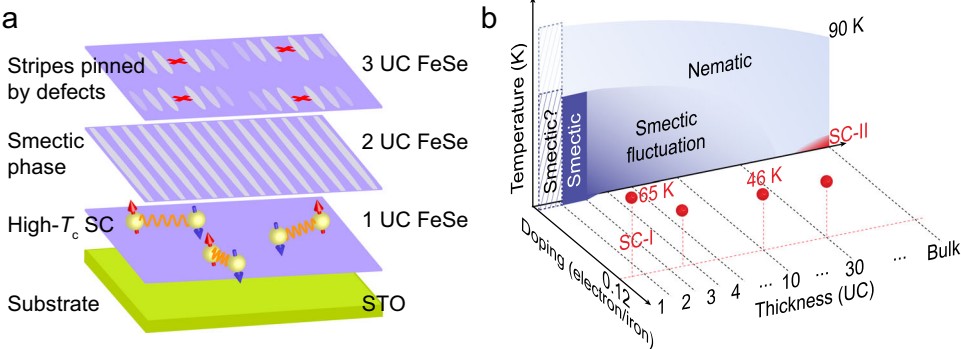

**Fig. 5 Phase diagram of FeSe/STO. a** Schematic of electronic structures in different layers of FeSe. The 1 UC FeSe/STO is a high-$T_c$ superconductor. Smectic phase with long-range stripe ordering is established in 2 UC FeSe films. 3 UC FeSe is in nematic phase with strong smectic fluctuation. Short-range stripes are pinned by defects. **b** Phase diagram of FeSe/STO as a function of temperature, thickness, and electron doping. The smectic phase may persist in 1 UC FeSe/STO once the electron doping is partially removed. In reality, by a large amount of electron doping from STO (0.12 electron/Fe), smectic phase is suppressed and high-$T_c$ superconductivity (SC-I) emerges in 1 UC FeSe, in which the residual smectic instability provides additional $T_c$ enhancement. In multilayer FeSe/STO, the superconducting states can be achieved by surface doping of alkali metals. The red balls denote the optimal $T_c$ obtained from ARPES results[38,39,52,53].

improved by Oxide-MBE[48]. The delicate effect of the strain warrants further theoretical and experimental investigations.

We now turn to discuss the microscopic origin of the smectic ordering. Since the stripe patterns are incommensurate, it is straightforward to relate them with the nesting of the Fermi surface. However, the smectic wave vector $q_0 = 0.19q_{Se}$ cannot match with any two bands in $k$-space of 2 UC FeSe (see details in Supplementary Note 12). Moreover, the $q_0$ shows thickness-independent behavior (if we compare it with that of the short-range stripes in multilayer FeSe film[21]). The thickness-independent wave vector is not expected under the Fermi surface nesting picture, because the band structures of FeSe are strongly correlated to the thickness of the films (see Supplementary Fig. 14)[39]. Instead, the strong electronic correlation needs to be considered. The electronic correlation in bulk FeSe is weaker than that in FeSe/STO, fully demonstrated in ARPES results by comparing the $d$-orbital band widths. Thus, even the short-range stripes are absent in bulk FeSe. While, in multilayer FeSe/STO with stronger strength of nematicity and correlation, short-range stripes are observed. Finally, in 2 UC FeSe/STO, which is at the strong limit of nematicity and electronic correlation, smectic phase emerges. We therefore demonstrate that the development of the long-range smectic phase derives from the largely enhanced electronic anisotropy and correlation in 2 UC FeSe (induced by STO substrate).

It is still an open question how to generate the correct smectic wave vector $q_0$ in FeSe/STO. Previous results demonstrate various frustrated spin fluctuations with different wave vectors of two-fold symmetry, such as $q(\pi, 0)$[49–51], $q(\pi, \pi/5)$[33], $q(\pi, \pi/3)$, $q(\pi, \pi/2)$, and other $q(\pi, Q < \pi/2)$[50], which compete with each other in FeSe. These competing commensurate wave vectors result in the phase separation of nematicity and magnetism in FeSe[50] but none of them can explain the development of the incommensurate stripe phase observed in our experiment. Very likely, the local moments[49,50], itinerant electrons[33], and orbital degree of freedom[34] collaborate together to generate the incommensurate smectic ordering.

The uniaxial strained LiFeAs shows similar smectic phase, in which incommensurate spin excitation is attributed as a possible origin[22]. However, it is still unclear whether such spin excitation also exists in 2 UC FeSe. In addition, to obtain the smectic phase, strains are introduced in both FeSe and LiFeAs, indicating the important role of phonon. But still, the strains in the two systems are quite different. In LiFeAs, external uniaxial stress, which naturally breaks rotational symmetry, is applied by piezo stacks.

While, in 2 UC FeSe, epitaxial strain is equivalent along the $a$- and $b$-directions. Thus, the smectic phase in FeSe emerges in the absence of external symmetry breaking.

We summarize the phase diagram of FeSe/STO as a function of temperature, thickness, and electron doping in Fig. 5b based on our findings and the previous experimental results[21,31,38,39,43,52,53]. The film thickness affects the electronic structure and controls the phase transition through the two parameters: lattice constant and doping concentration. The lattice expansion of FeSe induced by the lattice mismatch with STO substrate exists in FeSe films even with the thickness of tens of atomic layers, while the electron doping induced by STO is only prominent in few-layer FeSe films. The influences give rise to abundant electronic states in FeSe thin films.

Considering the geometry of FeSe/STO, the newly observed smectic phase is physically located in between the high-$T_c$ superconducting phase (1 UC FeSe with electron doping) and nematic phase (3 UC and thicker FeSe). Two smectic phase transitions occur in 3 UC and 1 UC FeSe, respectively. On the multilayer film (nematic phase) side, the short-range stripes (smectic fluctuations stabilized by defects) compete with superconductivity and lead to a wide-range non-superconducting phase[21,39]. Low-$T_c$ superconductivity of FeSe (SC-II region in the phase diagram) reemerges as lattice constant relaxes to the bulk value[43]. On the other side, as shown in the phase diagram, stronger smectic phase may persist in 1 UC FeSe with little electron doping. Along the doping-axis, in 1 UC FeSe, the smectic and nematic phases are suppressed by carrier transfer from STO, but the smectic electronic instability is still reserved, which can provide additional enhancement of superconductivity with optimal doping level of 0.12 electron/Fe (SC-I region). Thicker FeSe thin films show similar surface doping behaviors, but with less superconductivity enhancement due to the decaying smecitc/nematic fluctuations (see Figs. 3f, 4g, and the red balls in Fig. 5b). This is consistent with a recent theoretical proposal that superconductivity can be enhanced near a smectic/nematic QCP regardless of the pairing mechanism and pairing symmetry[31]. Therefore, besides the widely studied effects of STO, smectic instability in FeSe itself is another key factor that needs to be considered for the $T_c$ enhancement in FeSe/STO.

## Methods

**Sample preparation.** FeSe thin films were grown by molecular beam epitaxy method. The Nb-doped (0.05 wt%) STO(001) substrates were degassed in ultra-high vacuum chamber (base pressure is better than $3 \times 10^{-10}$ Torr) at 500 °C for several hours and subsequently annealed at 1150 °C for 20 min to obtain $TiO_2$

terminated surface. Then, high purity Fe (99.995%) and Se (99.9999%) sources were co-evaporated by two Knudsen cells to grow FeSe films on STO. During growth, the substrates were kept at 430 °C by applying DC current. The as-grown samples were annealed at 480 °C for hours to improve sample quality. The Rb deposition was performed in situ by using a rubidium dispenser (SAES Getters).

**STM measurements**. In situ STM measurements were carried out at 4.2 K in a commercial low-temperature STM (Unisoku). A polycrystalline PtIr STM tip was used and calibrated using Ag island before STM experiments. STS data were taken by standard lock-in method. The feedback loop is disrupted during data acquisition and the frequency of oscillation signal is 973.0 Hz.

## Data availability

The data that support the findings of this study are included in this article and its supplementary information file and are available from the corresponding author upon reasonable request.

## Code availability

The computer code used for data analysis is available upon request from the corresponding author.

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

## Acknowledgements

We thank Dung-Hai Lee, Yayu Wang, Hong Yao, and Zheng Liu for inspiring discussions. The experiments were supported by the Ministry of Science and Technology of China (No. 2016YFA0301002), the National Science Foundation (No. 11674191), and the

Beijing Advanced Innovation Center for Future Chip (ICFC). W.L. was also supported by Beijing Young Talents Plan and the National Thousand-Young-Talents Program.

## Author contributions
W.L. and Q-K.X. conceived and supervised the research project. Y.Y., X. F., and X.W. performed the STM experiments and grew the samples. W.L., Y.Y., X. F., Y.Z., and K.H. analyzed the data. W.L. and Y.Y. wrote the manuscript with input from all other authors.

## Competing interests
The authors declare no competing interests.
