## [Peer Review File · Nature Communications]

REVIEWER COMMENTS

Reviewer #1 (Remarks to the Author):

The manuscript by Yuan and collaborators describes smectic electronic states in thin films of FeSe on SrTiO₃ close to the single layer limit. The key discovery is the observation of these smectic electronic states in 2 unit cell thick films, with the smectic phase being suppressed both in thicker films as well as in the monolayer limit. The authors deduce a phase diagram of nematic phase, smectic phase and superconductivity.

The discussion of possible origins of the smectic modulation remains largely at a qualitative level. Is the modulation the authors observe linked to specific features of the Fermi surface? How does the electronic structure change as a function of thickness of FeSe? A direct comparison with the Fermi surface from ARPES (ref. 38) would be useful.

I find the manuscript interesting, and it provides new evidence for symmetry breaking in thin films of FeSe. The discussion remains somewhat vague about possible origins, which makes it difficult to assess the importance of the work. If this was stronger, and the other points mentioned below are addressed, the manuscript may become suitable for publication in Nature Communications. As is, I would rather see it in a more specialized journal.

A few minor comments:

- Static order should not be called fluctuation (line 26 of main text and section 3 of supplementary): the authors do not show any time-resolved data, so without evidence for a time dependence, one would expect this to be a static modulation of the charge density rather than a fluctuation.
- The authors may want to discuss the possible role of epitaxial strain in the formation of the smectic order. The wave vector the authors observe is surprisingly similar to the one seen in LiFeAs with uniaxial strain in ref. 22. It would be good to see a bit more discussion of this relation.

Reviewer #2 (Remarks to the Author):

Yuan et al., here report a STM study on the smectic phase in the few monolayers - FeSe film grown on the Nb-SrTiO₃ substrate. The enhanced high T_c superconductivity in one-unit cell (UC) FeSe film represents one of the most fascinating problems in condensed matter physics. Meanwhile, the relationship between electronic liquid crystal phase and high T_c superconductivity can be a key to unconventional Cooper pairing. Therefore, the subject of this manuscript is certainly important and suitable for the wide readership of Nature Communications.

The authors report the observation of an incommensurate smectic phase in a 2-UC FeSe sample, whose wavelength is relatively energy independent. The authors also report that the systems undergo a smectic-to-nematic quantum phase transition from 2UC to 3UC by comparing the STM images of 1UC, 2UC and 3UC samples. Furthermore, the authors show the smectic phase in the 2UC-FeSe "melts" at the higher temperature and they further construct a phase diagram based on these observations. The data presented here are of high quality and the discussion is very interesting; however, the authors should clarify several issues before I can recommend for publications.

(1) The authors stated that the nematic electronic structure persists at the elevated temperature in the 2UC-FeSe sample in page 5. They also marked the nematic phase at high temperature for the 2UC sample. However, no data is presented in the manuscript to demonstrate the nematic pattern exists in the 2UC sample. Even for the nematic data of the 3UC-FeSe sample shown in the SI, it is not clear how the nematic domain boundaries are determined although the short range striped around impurities rotate at the different domains.

(2) In the Fig.3c, the doping axis does not appear to offer much information regarding to its effect on the smectic or nematic phase. The doping level for the red line should be labelled clearly.

- (3) The stripes show a $\sim\pi$ phase shift from 30meV to 60meV in Fig.1d. Does this correlate with the electronic band structure?
- (4) I'd like to see the raw dI/dV data of Fig. 1d shown in the supplementary information.
- (5) No orange dashed lines in the supplementary figure 2.

Reviewer #3 (Remarks to the Author):

The authors used low-temperature scanning tunneling microscopy to study FeSe thin films grown on Nb-doped SrTiO₃ substrate. They found a smectic phase (which manifests itself as long-stripe order) in the 2 u.c. thick FeSe films, but this phase is absent in 1 u.c. and thicker FeSe films. The authors claimed that the stabilization of the smectic phase in the 2 u.c. FeSe films is due to a delicate balance between tensile strain and charge transfer. In the 1 u.c. FeSe films, charge transfer dominates and the smectic phase is subdued by superconductivity. In thicker films, the tensile strain gets weaker and the smectic phase loses the translational order and becomes a nematic phase.

The experimental results are solid and the discovery of the smectic phase in the 2 u.c. FeSe films deserves publication in some forms. However, in my opinion, the weakest point in this work is that the current experimental data are not sufficient to support the general claim in the paper.

1. The authors wrote in the paper that the smectic phase can also emerge once the dopants are appropriately removed from 1 u.c. FeSe/STO. If the authors grow 1 u.c. FeSe films on bare SrTiO₃ or other TiO₂-terminated insulating substrates (instead of Nb-doped SrTiO₃), could the authors reduce the charge transfer and stabilize the smectic phase in 1 u.c. FeSe films?

2. The authors wrote in the paper that the development of the long-range smectic phase derives from the largely enhanced electronic anisotropy and correlation in 2 u.c. FeSe (induced by SrTiO₃ substrate). What will happen if the authors grow FeSe films on Nb-doped BaTiO₃ (Nature Communications volume 5, Article number: 5044 (2014))? Since the lattice constant of BaTiO₃ is larger than that of SrTiO₃, shall we expect that the smectic phase can be stabilized in thicker FeSe films (not only 2 u.c.)?

3. The authors claimed that the residual smectic instability provides additional superconductivity T_c enhancement in the 1 u.c. FeSe films. On the other hand, the authors also mentioned that in multilayer FeSe/STO, the superconducting states can be achieved by surface doping of alkali metals. In 3 u.c. FeSe films, which are also in the vicinity of a smectic phase and the smectic fluctuations are strong, by using the surface doping of alkali metals (such as K), could the authors stabilize a superconducting phase in 3 u.c. FeSe films on SrTiO₃ substrate?

4. The authors wrote in the paper that the tensile strain in 2 u.c. FeSe is larger than that in 3 u.c. FeSe films. Usually, below some critical thickness, thin films are coherently strained by the substrate, i.e. the in-plane lattice constant of the films is identical to that of the substrate. So what is the critical thickness of FeSe grown on SrTiO₃ substrate (for the coherent strain)? If the critical thickness is larger than 3 u.c., could the

authors explain what they mean by "the tensile strain in 2 u.c. FeSe is larger than that in 3 u.c. FeSe films"? I would naively think the tensile strain is almost the same (because the lattice mismatch is the same).

5. This is a minor point: the authors used the phrase "quantum phase transition between 1 u.c. and 2 u.c. etc.". Usually, quantum phase transition is a **second-order** phase transition at zero temperature driven by quantum fluctuations. It is tuned by a continuous parameter. Here the tuning parameter is the thickness of FeSe films, which is discrete. Just to avoid unnecessary misunderstanding, I would suggest that the author use "a phase transition" rather than "a quantum phase transition".

Overall the discovery in this work is interesting and could be important for the understanding of Fe-based superconductivity. If the authors could perform additional experiments and provide convincing evidence to support the underlying physics that is claimed in the paper, I would be happy to recommend the publication of this work in Nature Communications.

REVIEWER COMMENTS

Reviewer #1 (Remarks to the Author):

The manuscript by Yuan and collaborators describes smectic electronic states in thin films of FeSe on SrTiO₃ close to the single layer limit. The key discovery is the observation of these smectic electronic states in 2 unit cell thick films, with the smectic phase being suppressed both in thicker films as well as in the monolayer limit. The authors deduce a phase diagram of nematic phase, smectic phase and superconductivity.

The discussion of possible origins of the smectic modulation remains largely at a qualitative level. Is the modulation the authors observe linked to specific features of the Fermi surface? How does the electronic structure change as a function of thickness of FeSe? A direct comparison with the Fermi surface from ARPES (ref. 38) would be useful.

I find the manuscript interesting, and it provides new evidence for symmetry breaking in thin films of FeSe. The discussion remains somewhat vague about possible origins, which makes it difficult to assess the importance of the work. If this was stronger, and the other points mentioned below are addressed, the manuscript may become suitable for publication in Nature Communications. As is, I would rather see it in a more specialized journal.

Reply: We appreciate that the reviewer thinks our manuscript is interesting. The reviewer gives us very nice suggestions which help us to improve the manuscript. The reviewer's most concerned point is the relationship between the stripe ordering, the band structures of FeSe films as well as the film's thickness. Accordingly, we carefully do more analysis to our data, carry out additional experiments and revise our manuscript. The corresponding results and discussions are added in the revised manuscript, Supplementary Information and also shown in the following point-by-point response. We believe all the reviewer's concerns are fully addressed.

1. Supplementary Figure 13 shows a direct comparison between the wave vectors of the stripe ordering and thickness dependent band structures of FeSe thin films measured by ARPES. Supplementary Fig. 13a is the FFT result in the inset of Fig. 1c. The wave vector length of the stripes (highlighted by red circles) is $q_0 = 0.309 \text{ \AA}^{-1}$. Supplementary Fig. 13b-f are thickness dependence of d_{yz} and d_{xy} bands around the M point extracted from ARPES data [*Nat. Mater.* 12, 634 (2013)]. The red arrow in Supplementary Fig. 13b denotes the wave vector of the stripe ordering, which cannot link to any two bands in k -space. Therefore, the itinerate electron picture, which attribute the origin of smectic phase to the scattering between two nesting bands, is excluded.

One may find that the d_{yz} and d_{xy} bands have two almost parallel branches near E_F , which

can give rise to a possible energy-independent inter-band scattering wave vector (the wave vector q_s denoted in black double headed arrows in Supplementary Fig. 13b-f). And it might lead to the energy-independent stripe ordering. However, this scattering process is inconsistent with the stripe ordering from the following three aspects: (1) In 2 UC FeSe thin film (Supplementary Fig. 13b), the stripe ordering wave vector q_0 is apparently shorter than the inter-band scattering wave vector $q_s = 0.406 \text{ \AA}^{-1}$. (2) The hole-like d_{yz} band reaches its band top within 60 meV above E_F , and such inter-band scattering above this energy could not occur anymore. However, the stripe ordering can still be observed at higher energy, for example at 150 meV (Fig. 2b and f in the main text). (3) The long-range stripe ordering in 2 UC FeSe and the short-range stripe ordering in 30 UC FeSe have the same period. In contrast, the inter-band scattering wave vector q_s is thickness-dependent and shorter in thicker films.

In short, the smectic phase in 2 UC FeSe cannot be explained by an itinerate electron picture and the electronic correlation effect needs to be considered. This part has been updated in the main text (the 1st Paragraph on Page 8) and Supplementary Note 11.

Supplementary Figure 13 Comparison of stripe ordering to band structure. **a** The FFT result in the inset of Fig 1c. The red circles highlight the scattering wave vector of the stripe ordering ($q_0 = 0.19q_{se} = 0.309 \text{ \AA}^{-1}$). The black square is the 2-Fe Brillouin zone boundary. **b-f** Thickness dependence of band structure around the M point extracted from ARPES results [Nat. Mater. 12, 634 (2013)]. The blue and green lines denote the d_{xy} and d_{yz} bands in multilayer FeSe. The double headed red arrow denotes the wave vector of the stripe ordering q_0 . The double headed black arrows denote the possible inter-band scattering wave vectors (q_s) between the parallel branches of d_{xy} and d_{yz} bands. The scattering wave vector q_s is thickness-dependent.

2. Regarding the role and importance of the smectic phase, we obtain more information based on our new experimental results. By carrying out surface doping experiments on 2 UC FeSe film, we find the smectic phase in 2 UC FeSe is gradually suppressed with electron doping of Rb atoms (Fig. 3) and superconductivity emerges as the smectic ordering is completely disappeared (Fig. 4). As shown in the revised Fig. 3 below, the long-range stripe is very sensitive to Rb surface dopants on the 2 UC FeSe surface. These new findings confirm and refine the phase diagram that we proposed in Fig. 5.

The details on this part have been updated in the revised Fig. 3 and its related discussions in the main text (the last paragraph on Page 5).

Fig. 3 Suppression of smectic phase in 2 UC FeSe by Rb surface doping. **a-e** Topographic images of 2 UC FeSe with surface Rb doping at various coverages: **a** 0 ML (30 nm \times 30 nm; set point, $V_s = 60$ mV, $I_t = 30$ pA), **b** 0.0060 ML (30 nm \times 30 nm; set point, $V_s = 60$ mV, $I_t = 50$ pA), **c** 0.0075 ML (30 nm \times 30 nm; set point, $V_s = 60$ mV, $I_t = 40$ pA), **d** 0.0135 ML (30 nm \times 30 nm; set point, $V_s = 100$ mV, $I_t = 20$ pA), **e** 0.0265 ML (30 nm \times 30 nm; set point, $V_s = 60$ mV, $I_t = 50$ pA). Inset of **b**: STM topographic image of a single Rb atom adsorbed on 2 UC FeSe (see Supplementary Figure 10). **f** Doping dependence of stripe area ratio (shown in blue dots) estimated from **a-e** (see Supplementary Figure 11). The blue and red shaded regions denote the smectic and superconducting states, respectively.

More importantly, by comparing the dopant-induced superconductivity of 2 UC and 3 UC FeSe (Fig. 4), we find the superconductivity of 2 UC FeSe is better than that of 3 UC FeSe. Given the fact that the main difference between 2 UC and 3 UC FeSe is whether the smectic phase exists or not, we demonstrate the importance of the smectic ordering: In 2 UC FeSe, the smectic phase is suppressed by electron doping, however, the residual smectic fluctuation provides additional enhancement of its superconductivity. The details on this part have been updated in Fig. 4 and its related discussions (Page 6 and Page 7).

A few minor comments:

- Static order should not be called fluctuation (line 26 of main text and section 3 of supplementary): the authors do not show any time-resolved data, so without evidence for a time dependence, one would expect this to be a static modulation of the charge density rather than a fluctuation.

Reply: Thank the reviewer for pointing it out. The description is indeed inappropriate. We think that the defects locally stabilize the fluctuation and form short-range stripe ordering. Therefore, the short-range stripe ordering reflects the existence of fluctuation but should not be called “fluctuation”. In the revised manuscript, we correct the “fluctuation” to “short-range stripe ordering” in line 26 of main text and Supplementary Note 6.

- The authors may want to discuss the possible role of epitaxial strain in the formation of the smectic order. The wave vector the authors observe is surprisingly similar to the one seen in LiFeAs with uniaxial strain in ref. 22. It would be good to see a bit more discussion of this relation.

Reply: The reviewer raises a very good point. The smectic phase in 2 UC FeSe shows several similarities to that in LiFeAs. For example, the wave vectors of the stripes in these two systems are both incommensurate and along the Fe-Fe direction. Those similarities indicate that the smectic orderings perhaps have the same origin. In LiFeAs, the incommensurate spin excitation is attributed as a possible origin for the smectic phase. But it is still unclear whether such spin excitation also exists in 2 UC FeSe. Currently, although we have put a lot of efforts, neutron scattering experiments on FeSe film cannot be carried out due to technical issues.

Strains are introduced in both FeSe and LiFeAs, indicating the importance of phonon for the development of smectic ordering. However, the strains in the two systems are also different. In LiFeAs, external uniaxial stress, which naturally breaks rotational symmetry, is applied by piezo stacks. While, in 2 UC FeSe, epitaxial strain is introduced by lattice mismatch between the substrate SrTiO₃ and FeSe thin film, in which the tensile strain is

equivalent along the a - and b -directions. Thus, the smectic phase in FeSe emerges in the absence of external symmetry breaking.

Following the reviewer's suggestion, we have added the discussion above to the main text (the 3rd paragraph of Page 8).

Reviewer #2 (Remarks to the Author):

Yuan et al., here report a STM study on the smectic phase in the few monolayers - FeSe film grown on the Nb-SrTiO₃ substrate. The enhanced high T_c superconductivity in one-unit cell (UC) FeSe film represents one of the most fascinating problems in condensed matter physics. Meanwhile, the relationship between electronic liquid crystal phase and high T_c superconductivity can be a key to unconventional Cooper pairing. Therefore, the subject of this manuscript is certainly important and suitable for the wide readership of Nature Communications.

The authors report the observation of an incommensurate smectic phase in a 2-UC FeSe sample, whose wavelength is relatively energy independent. The authors also report that the systems undergo a smectic-to-nematic quantum phase transition from 2UC to 3UC by comparing the STM images of 1UC, 2UC and 3UC samples. Furthermore, the authors show the smectic phase in the 2UC-FeSe “melts” at the higher temperature and they further construct a phase diagram based on these observations. The data presented here are of high quality and the discussion is very interesting; however, the authors should clarify several issues before I can recommend for publications.

Reply: We thank the reviewer for thinking our data are of high quality and the results are of interests to the wide readership. Based on the reviewer’s nice suggestions, we carry out more experiments and carefully revise our manuscript. The reviewer’s questions are responded in the following point-by-point reply. We believe all the reviewer’s concerns are fully addressed.

(1) The authors stated that the nematic electronic structure persists at the elevated temperature in the 2UC-FeSe sample in page 5. They also marked the nematic phase at high temperature for the 2UC sample. However, no data is presented in the manuscript to demonstrate the nematic pattern exists in the 2UC sample. Even for the nematic data of the 3UC-FeSe sample shown in the SI, it is not clear how the nematic domain boundaries are determined although the short range striped around impurities rotate at the different domains.

Reply: The existence of nematic phase in 2 UC FeSe at elevated temperature has already been demonstrated in ARPES results [*Nat. Mater.* 12, 634 (2013); *PRB.* 94, 115153 (2016)].

The signature for nematicity in FeSe is the splitting of d_{xz} and d_{yz} bands. Temperature-dependent ARPES results show such band splitting in FeSe films with the thickness ≥ 2 UC. Moreover, the nematic transition temperature increases from 120 K to 170 K when the thickness of the film decreases from 30 to 2 UC, indicating enhanced anisotropy

in thinner FeSe film. The ARPES results also support our picture: electronic correlation is stronger in thinner films and finally induces smectic phase transition in 2 UC FeSe.

In STM measurements, the d_{xz} and d_{yz} band splitting of FeSe thin films manifests as unidirectional quasiparticle interference patterns [*Nat. Phys.* 13, 957 (2017); *Nat. Commun.* 12, 10 (2021)].

To further demonstrate the nematicity of FeSe film in our study, we perform additional QPI measurements in 3 UC and 2 UC FeSe to investigate the nematicity induced symmetry breaking. The QPI patterns of 2 UC FeSe are also shown below. The dispersive QPI features are more clearly shown in Supplementary Movie 1-4.

Supplementary Figure 5 and 6 show the QPI results taken on a 2 UC FeSe film at positive and negative energies, respectively. Two defects are introduced as scattering centers (Supplementary Figure 5a), in which long-range stripes are clearly revealed. Above E_F , the unidirectional QPI patterns propagate along the direction of the long-range stripes (b -direction), which break C_4 symmetry (see the two magenta arrows in Supplementary Figure 5b-k). The patterns move towards the scattering centers at higher energies, indicating a shorter wavelength in real-space (or a larger scattering wave vector in q -space). Such an energy dispersive behavior corresponds to an electron-like band above E_F . More importantly, such scattering pattern is absent along the a -direction, reflecting an inequivalent band structure along the a - and b - directions.

At negative energies, the QPI results also exhibit unidirectional patterns (see the yellow arrows in Supplementary Figure 6), but they are along the a -direction (perpendicular to the patterns taken above E_F). The QPI patterns show two ring-like features in the vicinity of defects (denoted by the yellow dashed rings in Supplementary Fig. 6a, b). The rings become larger with the energy closer to E_F , reflecting a scattering process from a hole-like band. The lack of such hole-like band feature along the b -direction again indicates the C_2 symmetry of the band structure. Similar QPI patterns are more clearly observed in 3 UC FeSe (Supplementary Figure 3 and 4), indicating 2 UC and 3 UC FeSe have similar band structures.

More importantly, the scattering wave vectors \mathbf{q} extracted from our QPI measurements perfectly match the band structures from ARPES measurements (see the yellow double-headed arrows in Supplementary Figure 7, also shown below). The details of analysis are presented in Supplementary Note 5. Based on the comparison between STM and ARPES results, we demonstrate that the anisotropic QPI patterns indeed originate from the nematicity induced inequivalent band structure in k -space. Therefore, ARPES and STM results consistently prove the existence of nematicity in 2 UC and 3 UC FeSe.

To make the manuscript clearer, we have added one sentence in the main text (the 2nd paragraph of Page 5). The corresponding results and detailed discussion are added to Supplementary Note 3-5.

Supplementary Figure 5 Quasiparticle interference in 2 UC FeSe at positive energy. **a** STM topographic image of two defects in 2 UC FeSe film (20 nm \times 16 nm; set point, $V_s = 60$ mV, $I_t = 200$ pA). **b-k** Positive energy dI/dV mappings taken at the same position (20 nm \times 16 nm; set point, $V_s = 120$ mV, $I_t = 500$ pA). The magenta arrows denote the energy-dependent QPI patterns. **l** Energy dispersion of the scattering wave vector q_b . A parabolic fitting result is shown in a blue curve.

Supplementary Figure 6 Quasiparticle interference in 2 UC FeSe at negative energy. a-h Negative energy dI/dV mappings taken at the same position ($20 \text{ nm} \times 16 \text{ nm}$; set point, $V_s = 120 \text{ mV}$, $I_t = 500 \text{ pA}$). The yellow arrows denote the energy-dependent QPI patterns. **i** Energy dispersion of the scattering wave vector q_a . A parabolic fitting result is shown in a blue curve.

Supplementary Figure 7 Band structure and QPI. **a** Schematic Fermi surface of FeSe thin film around the M point [*Nat. Mater.* 12, 634 (2013); *PRB.* 94, 115153 (2016)]. **b, c** Schematic band structures along the cut 1 and cut 2 in **a**, respectively [*PRB.* 94, 115153 (2016)]. The double headed yellow arrow denotes the inter-orbital scattering process between the d_{yz} and d_{xy} bands below E_F . The double headed magenta arrow denotes the intra-orbital scattering process of d_{yz} band above E_F . **d, e** d_{yz} and d_{xy} band structures around the M point of 2 UC and 3 UC FeSe thin films extracted from ARPES results [*Nat. Mater.* 12, 634 (2013)]. The double headed yellow arrows are the scattering wave vectors obtained from QPI data in Supplementary Figure 4 and 6.

Regarding the reviewer's question on how to determine the nematic domain boundaries in 3 UC FeSe, we can distinguish them in STM topographic image with various bias voltages.

In STM image, the nematic domain boundaries can be clearly imaged and they can be in opposite contrasts at different scanning bias voltages [see Fig. 2a and b in *Nano Lett.* 18, 7176 (2018)]. The apparent height of the domain boundaries is lower than that of domain regions at positive bias (0 mV to 100 mV), while it reverses at negative bias (-100 mV to 0 mV). Therefore, the nematic domain boundaries in Supplementary Figure 8, which manifests in lower apparent height, are easily distinguished.

As the reviewer mentioned, the orientation of the stripes is another benchmark to search the nematic domains. Based on our previous results [*Nat. Phys.* 13, 957 (2017); *Nano Lett.* 18, 7176 (2018)], the relative orientations of the stripes and the dispersive QPI patterns are always locked together (also see in Supplementary Figure 3 and 4), therefore an easy way to

find the nematic domains is to check the orientation of the short-range stripes directly (to check if they are perpendicular to each other).

(2) In the Fig.3c, the doping axis does not appear to offer much information regarding to its effect on the smectic or nematic phase. The doping level for the red line should be labelled clearly.

Reply: The reviewer raises a very good point. The study of doping effect on the smectic phase can provide further information for its origin and relationship to superconductivity. We therefore employ additional experiments to systematically study the doping dependence of smecticity. Rb atoms are deposited as dopants to the surface of 2 UC and 3UC FeSe films. Long-range stripe orderings in 2 UC FeSe are suppressed with the increased surface Rb concentration and then superconductivity emerges. We add the corresponding results in the revised manuscript (Fig. 3a-f and Supplementary Note 9) and also present below.

Figure 3 demonstrates the topographic images of 2 UC FeSe with various Rb coverages. The Rb atoms can locally suppress the stripe orderings. To be specific, the long-range stripe orderings apparently detour around the Rb atoms, which leads to a dramatic phase decoherence of the stripes in the regions with high Rb density (Fig. 3b). In the low Rb density regions, the stripes are still of long-range coherence. As shown in Fig. 3c and d, when the overall doping concentration increases, the area with stripe patterns gradually reduces. When the Rb coverage increases to 0.0265 ML, the stripe ordering is totally suppressed (Fig. 3e). Fig. 3f summarizes the decrease of stripes as the Rb coverage increases, in which the ratios of stripe area are estimated from the topographic images (Fig. 3a-e and Supplementary Note 9).

Fig. 3 Suppression of smectic phase in 2 UC FeSe by Rb surface doping. **a-e** Topographic images of 2 UC FeSe with surface Rb doping at various coverages: **a** 0 ML (30 nm \times 30 nm; set point, $V_s = 60$ mV, $I_t = 30$ pA), **b** 0.0060 ML (30 nm \times 30 nm; set point, $V_s = 60$ mV, $I_t = 50$ pA), **c** 0.0075 ML (30 nm \times 30 nm; set point, $V_s = 60$ mV, $I_t = 40$ pA), **d** 0.0135 ML (30 nm \times 30 nm; set point, $V_s = 100$ mV, $I_t = 20$ pA), **e** 0.0265 ML (30 nm \times 30 nm; set point, $V_s = 60$ mV, $I_t = 50$ pA). Inset of **b**: STM topographic image of a single Rb atom adsorbed on 2 UC FeSe (see Supplementary Figure 10). **f** Doping dependence of stripe area ratio (shown in blue dots) estimated from **a-e** (see Supplementary Figure 11). The blue and red shaded regions denote the smectic and superconducting states, respectively.

Actually, superconductivity in 2 UC FeSe starts to emerge at the Rb coverage of 0.0265 ML. Therefore, we also carry out systematic study of the Rb doping effect on superconductivity in 2 UC and 3 UC FeSe films (Fig. 4), and the details are added in the revised manuscript.

We thank the reviewer for reminding us the doping level in the phase diagram is not clearly labelled. It has been enlarged in the revised manuscript (Fig. 5b and its Figure Caption).

(3) The stripes show a $\sim\pi$ phase shift from 30meV to 60meV in Fig.1d. Does this correlate with the electronic band structure?

Reply: We have carefully compared the wave vector of stripe ordering and the electronic band structure of FeSe (Supplementary Note 11) and find they are irrelevant. Therefore, the itinerate electron picture cannot explain the stripe phase here, and the π phase shift of the stripes is not correlated with band structure.

Interestingly, previous study on uniaxial strained LiFeAs also found similar π phase shift of stripe orderings within a small energy range [*Nat. Commun.* 9, 2602 (2018)]. But its origin is also unclear.

(4) I'd like to see the raw dI/dV data of Fig. 1d shown in the supplementary information.

Reply: The raw dI/dV data of Fig. 1d is added into the Supplementary Note 2 and also presented below for clarity.

Supplementary Fig. 2a shows the raw dI/dV data in Fig. 2d. The spectra in different colors correspond to the different energy ranges in Fig. 1d. The gray dashed lines highlight the peak and dip positions of the stripe ordering. It is hard to make a direct analysis for these raw data, because the oscillation amplitudes of the spectra are significant different from each other. In order to give a complementary analysis other than calculating the d^3I/dV^3 that is shown in the main text, each spectrum is divided by its linear background, re-scaled by multiplying a factor, shifted for clarity and finally shown in Supplementary Fig. 2b (the corresponding energies and scaling factors are labeled on the right of the spectra). It shows that the period of the stripes doesn't change with energy and the stripes in 30 meV to 60 meV have a π phase shift. These conclusions are consistent with those obtained from the d^3I/dV^3 spectra in Fig. 2d.

Supplementary Figure 2 dI/dV line-cut in Fig. 1d. **a** Raw dI/dV spectra. **b** The normalized

dI/dV spectra. The spectra are shifted for clarity. The red, green and blue colors correspond to the different energy ranges in Fig. 1d. The gray dashed lines denote the peak and dip positions in the spectra.

(5) No orange dashed lines in the supplementary figure 2.

Reply: We thank the reviewer for reminding us the orange dashed line is not clear. We have revised the figure in Supplementary Figure 8.

Reviewer #3 (Remarks to the Author):

The authors used low-temperature scanning tunneling microscopy to study FeSe thin films grown on Nb-doped SrTiO₃ substrate. They found a smectic phase (which manifests itself as long-stripe order) in the 2 u.c. thick FeSe films, but this phase is absent in 1 u.c. and thicker FeSe films. The authors claimed that the stabilization of the smectic phase in the 2 u.c. FeSe films is due to a delicate balance between tensile strain and charge transfer. In the 1 u.c. FeSe films, charge transfer dominates and the smectic phase is subdued by superconductivity. In thicker films, the tensile strain gets weaker and the smectic phase loses the translational order and becomes a nematic phase.

The experimental results are solid and the discovery of the smectic phase in the 2 u.c. FeSe films deserves publication in some forms. However, in my opinion, the weakest point in this work is that the current experimental data are not sufficient to support the general claim in the paper.

Reply: We really appreciate that the reviewer thinks our experimental results are solid and deserve for publication. We thank the reviewer for the constructive suggestions, which help us a lot to improve the manuscript. Based on these nice comments, we carry out more experiments, do more analysis and revise the manuscript. All the reviewer's concerns are addressed in the following point-by-point response. We believe the revised manuscript is suitable for publication in Nature Communications.

1. The authors wrote in the paper that the smectic phase can also emerge once the dopants are appropriately removed from 1 u.c. FeSe/STO. If the authors grow 1 u.c. FeSe films on bare SrTiO₃ or other TiO₂-terminated insulating substrates (instead of Nb-doped SrTiO₃), could the authors reduce the charge transfer and stabilize the smectic phase in 1 u.c. FeSe films?

Reply: The reviewer raises a very nice point. We are always trying to conduct such kind of experiments. It can provide important information for understanding the superconductivity of 1 UC FeSe/STO if we can reduce the charge transfer from the substrate.

However, the charge transfer from STO to FeSe is robust. Using bare SrTiO₃ or TiO₂-terminated insulating substrates cannot reduce the charge transfer, because the charge transfer is governed by the imbalance of the work functions between STO/TiO₂ and FeSe. Those substrates (with no dopants) are still able to provide enough charge transfer to FeSe, which has been directly demonstrated by STM, UPS and EELS experiments [*Phys. Rev. Lett.* **117**, 067001 (2016); *Nat. Commun.* **8**, 214 (2017); *Sci. Adv.* **4**, eaao2682 (2018)].

In a previous transport measurement [*Phys. Rev. B* 89, 060506(R) (2014)], the transition temperature of FeSe/STO only shows a variation of ~ 10 K after applying -200 V gate voltage (higher gate voltage will breakdown the substrate) and it can never reach the non-superconducting regime. Other stronger tuning methods, like ion-liquid or ion-solid gating, can hardly be combined with STM.

Although we cannot reduce the charge transfer in 1 UC FeSe, instead, as shown in the revised manuscript, we increase the electron doping level of 2 UC FeSe via surface deposition of Rb atoms. Systematic evolution of phase transition from smecticity to superconductivity are obtained. Those new results further support our conclusion. The details of the effects of doping are shown in the response to the reviewer's 3rd question and also updated in the revised Fig. 3 and related discussions in the main text. We hope the reviewer thinks that the additional experimental results are sufficient to support our claim now.

2. The authors wrote in the paper that the development of the long-range smectic phase derives from the largely enhanced electronic anisotropy and correlation in 2 u.c. FeSe (induced by SrTiO₃ substrate). What will happen if the authors grow FeSe films on Nb-doped BaTiO₃ (Nature Communications volume 5, Article number: 5044 (2014))? Since the lattice constant of BaTiO₃ is larger than that of SrTiO₃, shall we expect that the smectic phase can be stabilized in thicker FeSe films (not only 2 u.c.)?

Reply: This is a very interesting idea. Accordingly, in the past several weeks, we have put a lot of efforts on the growth of FeSe on BaTiO₃. We have to admit that the growth of FeSe on BaTiO₃ (BTO) is very challenging.

Firstly, we could not obtain any Nb-doped BaTiO₃ single crystals, and instead, we bought some bare BTO to do the growth. However, bare BTO is not conductive and cannot be measured by low-temperature STM; STM is essential for the study of the stripes. Based on our experiences on the STM study of FeSe/bare-STO, the key point relies on the precise control of the density of oxygen vacancies in the substrate: FeSe thin film cannot crystalize on STO with too much oxygen vacancies, while low-temperature STM measurement cannot be conducted if the concentration of oxygen vacancies is too low. Fortunately, we found the narrow window of the density of oxygen vacancy in STO, which can simultaneously guarantee the FeSe growth and STM measurement.

Unfortunately, the narrow window doesn't exist in bare BTO based on our experimental results. In order to precisely generate oxygen vacancies, BaTiO₃ substrate is annealed in UHV chamber with temperature increases step by step (from 600 °C as a starting point). Each annealing step holds for 30 min and then BTO substrate is transferred into STM to check its conductivity. We find that BTO is too insulating to carry out STM measurement until the

annealing temperature rises to 1000 °C. However, after annealing at this temperature, the color of the substrate changes to black due to the large amount of oxygen vacancies. This shows a sharp contrast to bare STO, which only requires a 600 °C annealing to generate enough oxygen vacancies to do STM measurement. Meanwhile, the color of STO is still white, indicating the low density of oxygen vacancies.

Figure R1a shows the topography of bare BaTiO₃ after annealing at 1000 °C. Terraces are formed with the height of 405 pm, corresponding to 1 UC BTO. A zoom-in image on the terrace (Fig. R1b) doesn't show 1 × 1 atomic resolution or surface reconstruction, and clusters are observed. The root-mean-square (RMS) surface roughness is 57 pm, which is not as flat as that of SrTiO₃, whose RMS roughness is usually less than 35 pm. After the treatment of BTO substrate, Fe and Se sources are co-evaporated with the substrate temperature of 430 °C. However, FeSe cannot crystalize on the BTO surface and only Fe clusters are formed, as shown in Fig. R1c, indicating the concentration of oxygen vacancies is too high to grow FeSe thin film. In order to control the concentration of oxygen vacancies more precisely, we also try to re-oxidize the substrate by annealing it with injecting ozone before FeSe growth. But we still cannot grow FeSe thin film on it.

Fig. R1 **a** STM topographic image of bare BaTiO₃ after annealing at 1000 °C (400 nm × 400 nm; set point, $V_s = 2.0$ V, $I_t = 20$ pA). **b** A zoom-in topographic image taken on a single terrace of bare BaTiO₃ (50 nm × 50 nm; set point, $V_s = 1.0$ V, $I_t = 50$ pA). **c** Topographic image taken on bare BaTiO₃ after growing FeSe (300 nm × 300 nm; set point, $V_s = 2.0$ V, $I_t = 20$ pA). Only Fe clusters are obtained.

Our experiment demonstrates that, compared with bare SrTiO₃, bare BaTiO₃ requires more oxygen vacancies to conduct STM measurement, on which FeSe film is not able to crystalize any more. The surface of treated BTO is also not as clean as that of STO. We suspect this is the reason why R. Peng *et al* use oxide-MBE growth of Nb-doped BaTiO₃ thin films to serve as substrates rather than BTO single crystals [*Nat. Commun.* 5, 5044 (2014)]. Oxide-MBE growth is very complicated, currently we could not do it in our lab.

We hope the reviewer could understand the challenges for the BTO-related experiments. We would like to emphasize that our claim, the enhanced electronic anisotropy and

correlation in 2 UC FeSe are important for the development of smectic phase, is based on solid experimental facts. The nematicity and correlation in single crystal FeSe is weaker than that in FeSe/STO, which has been fully demonstrated in ARPES results by comparing the d_{yz} and d_{xz} band splitting sizes and the d -orbital band widths. Therefore, even the short-range stripes are absent in single crystal FeSe. While, in multilayer FeSe films on STO, the nematicity and correlation is stronger, giving rise to the short-range stripes induced by defects. Finally, in 2 UC FeSe film on STO, which is actually at the strong limit of nematicity and electronic correlation (with the largest band splitting and the smallest d -orbital band width in FeSe system), smectic phase emerges. To make our point clearer, we have added the discussions above to the main text (the 1st paragraph of Page 8).

In addition, in the revised manuscript, we focus on another key parameter, electron doping, which plays significant role for smectic phase and superconductivity as well. We carry out additional experiments to study the doping effect on the smectic phase in FeSe. With the tuning knob of Rb surface doping, we can systematically control the transition of the smectic phase in 2 UC FeSe. The details are shown in the revised Fig. 3 and its corresponding discussions in the main text. The doping effects on the smectic phase and superconductivity in 2 UC and 3 UC FeSe will also be elaborated below in the response for the next question as well.

3. The authors claimed that the residual smectic instability provides additional superconductivity T_c enhancement in the 1 u.c. FeSe films. On the other hand, the authors also mentioned that in multilayer FeSe/STO, the superconducting states can be achieved by surface doping of alkali metals. In 3 u.c. FeSe films, which are also in the vicinity of a smectic phase and the smectic fluctuations are strong, by using the surface doping of alkali metals (such as K), could the authors stabilize a superconducting phase in 3 u.c. FeSe films on SrTiO₃ substrate?

Reply: This is a very nice comment. Yes, we can stabilize the superconductivity in 2 UC and 3 UC FeSe films on STO by Rb surface doping. Based on the reviewer's nice suggestion, we have carried out systematic surface doping experiments on FeSe thin film and investigate their influence on the stripe orderings and superconductivity. These new results further support our conclusion. The corresponding results (revised Fig. 3 and Fig. 4) and discussions are added in the revised manuscript (Page 5-7) and also presented below.

Figure 3 demonstrates the topographic images of 2 UC FeSe with various Rb coverages. The Rb atoms can locally suppress the stripe orderings. To be specific, as shown in Fig. 3b, the long-range stripe orderings apparently detour around the Rb atoms, which leads to a dramatic phase decoherence of the stripes in the regions with high Rb concentration. In the low Rb concentration regions, the stripes are still of long-range coherence. When the overall

doping concentration increases, as shown in Fig. 3c and d, the stripe area gradually reduces. When the Rb coverage increases to 0.0265 ML, the stripe ordering is totally suppressed (Fig. 3e). Fig. 3f summarizes the decrease of stripes as the Rb coverage increases, in which the ratios of stripe area are estimated from the topographic images (Fig. 3a-e and Supplementary Figure 11).

The doping level of 1 UC FeSe/STO is estimated to be 0.12 electron/Fe. The long-range stripe ordering is therefore absent at such a high doping level.

Fig. 3 Suppression of smectic phase in 2 UC FeSe by Rb surface doping. **a-e** Topographic images of 2 UC FeSe with surface Rb doping at various coverages: **a** 0 ML (30 nm × 30 nm; set point, $V_s = 60$ mV, $I_t = 30$ pA), **b** 0.0060 ML (30 nm × 30 nm; set point, $V_s = 60$ mV, $I_t = 50$ pA), **c** 0.0075 ML (30 nm × 30 nm; set point, $V_s = 60$ mV, $I_t = 40$ pA), **d** 0.0135 ML (30 nm × 30 nm; set point, $V_s = 100$ mV, $I_t = 20$ pA), **e** 0.0265 ML (30 nm × 30 nm; set point, $V_s = 60$ mV, $I_t = 50$ pA). Inset of **b**: STM topographic image of a single Rb atom adsorbed on 2 UC FeSe (see Supplementary Figure 10). **f** Doping dependence of stripe area ratio (shown in blue dots) estimated from **a-e** (see Supplementary Figure 11). The blue and red shaded regions denote the smectic and superconducting states, respectively.

Actually, superconductivity in 2 UC FeSe starts to emerge at the Rb coverage of 0.0265 ML. We also carry out systematic study of the Rb doping effect on superconductivity in 2 UC and 3 UC FeSe films.

Figure 4 summarizes the doping dependence of superconducting gaps in 2 UC and 3 UC FeSe thin films. In our experiment, we randomly take 100 dI/dV spectra at each doping level of 2 UC and 3 UC FeSe. We find that superconducting gap presents inhomogeneity and the strength of inhomogeneity varies with doping concentrations. Therefore, the superconductivity in Rb-coated FeSe films are characterized by two aspects: the homogeneity and the gap size.

The dI/dV spectra taken at each doping level are sorted into two groups on the basis of their superconducting gaps. The good superconducting group contains the spectra that show clean and symmetric (with E_F) superconducting gaps. The averaged spectra for good superconducting groups in 2 UC and 3 UC FeSe are shown in Fig. 4a and Fig. 4c, respectively. The bad superconducting group includes the spectra that show asymmetric superconducting gaps or contain in-gap states. The averaged spectra for bad superconducting groups in 2 UC and 3 UC FeSe are shown in Fig. 4b and Fig. 4d, respectively. In 2 UC FeSe, superconductivity emerges with 0.0265 ML Rb doping (Fig. 4a). While it doesn't appear in 3 UC FeSe until the Rb coverage increases to 0.038 ML (Fig. 4c).

Here, we define a good-superconductivity ratio as the proportion of the dI/dV spectra showing good superconducting gap at each Rb coverage. The good-superconductivity ratios are counted and shown in Fig. 4a and Fig. 4c at each doping level. The Rb coverage-dependent ratio curves of 2 UC and 3 UC FeSe are summarized in Fig. 4e. In both 2 UC and 3 UC FeSe thin films, the ratio increases with Rb coverage at the beginning and finally drops in the over-doped regime, exhibiting a dome-like feature. Comparing with 3 UC FeSe, 2 UC FeSe presents a higher maximum value of the homogeneity ratio (94.1% in 2 UC vs. 90.9% in 3 UC, also see the grey dashed guide line in Fig. 4e). In addition, the ratio curve of 2 UC FeSe shows a terrace-like shape, indicating that it has a wider doping range with high homogeneity of superconductivity. The maximum values of the ratios for 2 UC and 3 UC FeSe correspond the Rb coverage of 0.095 ML and 0.145 ML, respectively, at which the two averaged dI/dV spectra show decent superconducting gap features (Fig. 4f). Compared with that of 3 UC FeSe, the averaged dI/dV spectrum of 2 UC FeSe possesses much sharper coherence peaks, indicating even better superconductivity.

On the other hand, the superconducting gap sizes of each dI/dV spectra of the good superconducting groups are extracted and taken into statistics. The doping dependence of gap size is shown in Fig. 4g. Superconductivity in 2 UC and 3 UC FeSe both present dome-like features. The gap size in optimally doped 2 UC FeSe (12.75 meV) is larger than that in 3 UC FeSe (11.56 meV), which again shows that superconductivity in 2 UC FeSe is better.

In summary, our new experimental results show that the surface doped alkali metals can suppress the smectic phase in 2 UC FeSe and then introduce superconductivity. These results

further support our conclusions: (1) The suppression of smectic phase by surface electron doping in 2 UC FeSe confirms and refines the phase diagram in Fig. 5. (2) By comparing the two key aspects of superconductivity that we mentioned before, the homogeneity and the gap size, we demonstrate that the dopant-induced superconductivity in 2 UC FeSe is better than that in 3 UC FeSe. Given the fact that the main difference between 2 UC and 3 UC FeSe is whether the long-range smectic phase exists, our claim on the relationship of superconductivity and smectic phase is effectively supported. The long-range stripe orderings compete with the superconductivity in undoped 2 UC FeSe. While as the stripes are suppressed by doping, the residual smectic fluctuations provide additional enhancement of superconductivity.

Fig. 4. Superconductivity induced by Rb surface doping in 2 UC and 3 UC FeSe thin films. a, b Doping dependence of averaged dI/dV spectra taken on 2 UC FeSe with good and bad superconducting gaps, respectively. **c, d** Doping dependence of averaged dI/dV spectra

taken on 3 UC FeSe with good and bad superconducting gaps, respectively. Here, symmetric coherent peaks and absence of in-gap state are the criterions for good superconducting gap. In opposite, the spectra which show asymmetric superconducting gap or in-gap states are sorted into the bad superconducting group. **e** The dependence of good-superconductivity ratio on Rb coverage. Here, the ratio is defined as the proportion of the dI/dV spectra showing good superconducting gap at each Rb coverage. The higher ratio indicates better homogeneity. The curves are guides to the eye. **f** Averaged dI/dV spectra extracted from **a** and **c** at the doping levels with the optimal homogeneity in 2 UC and 3 UC FeSe. **g** Dependence of the superconducting gap size on Rb coverage. The spectra in good superconducting group are taken into statistics. The error bars denote the standard deviations of superconducting gap sizes at different Rb coverages.

4. The authors wrote in the paper that the tensile strain in 2 u.c. FeSe is larger than that in 3 u.c. FeSe films. Usually, below some critical thickness, thin films are coherently strained by the substrate, i.e. the in-plane lattice constant of the films is identical to that of the substrate. So what is the critical thickness of FeSe grown on SrTiO₃ substrate (for the coherent strain)? If the critical thickness is larger than 3 u.c., could the authors explain what they mean by "the tensile strain in 2 u.c. FeSe is larger than that in 3 u.c. FeSe films"? I would naively think the tensile strain is almost the same (because the lattice mismatch is the same).

Reply: In FeSe/SrTiO₃, the tensile strain is gradually relaxed rather than coherently persist to a critical thickness. Previous ARPES measurement found that the Brillouin zone of FeSe expands with film thickness increases (from 1 UC), indicating the gradual relaxation of the lattice [*Nat. Mater.* 12, 634 (2013)].

Usually, the coherent strain exists in thin films with 3D lattice structure, in which the coupling along the z -axis is strong. The FeSe thin film, however, is a 2D layered material whose inter-layer coupling is relatively weak. That is the reason why the strain in FeSe thin film is not coherent.

5. This is a minor point: the authors used the phrase "quantum phase transition between 1 u.c. and 2 u.c. etc.". Usually, quantum phase transition is a *second-order* phase transition at zero temperature driven by quantum fluctuations. It is tuned by a continuous parameter. Here the tuning parameter is the thickness of FeSe films, which is discrete. Just to avoid unnecessary misunderstanding, I would suggest that the author use "a phase transition" rather than "a quantum phase transition".

Reply: We thank the reviewer for the nice suggestion. We thought the driven force of the smectic phase is lattice constant or strain, which is continuous. That's why we use "quantum phase transition". But indeed, we cannot continuously control this parameter in real

experiment. To avoid unnecessary misunderstanding, we use “phase transition” in the revised manuscript.

Overall the discovery in this work is interesting and could be important for the understanding of Fe-based superconductivity. If the authors could perform additional experiments and provide convincing evidence to support the underlying physics that is claimed in the paper, I would be happy to recommend the publication of this work in Nature Communications.

REVIEWERS' COMMENTS

Reviewer #1 (Remarks to the Author):

The authors have made a good effort to address my comments and included significantly more data to support their conclusions.

There is one aspect in the new section which I think needs clarification: In the new section, the authors discuss "good" and "bad" superconducting gap spectra - this should be qualified better. Is the selection of "good" and "bad" spectra done by eye, or by an algorithm? In either case, the criteria for a spectrum to be classed as "good" or "bad" need to be stated clearly, either in the main text or in the supplementary material - as otherwise it appears rather subjective. For example for asymmetry of the gap, mentioned as a criterion, does this refer to the gap not having the same gap magnitude at positive and negative bias, the asymmetry of the coherence peaks (or gap edge), or both?

Once this point is addressed, I would support publication of the revised manuscript.

Reviewer #2 (Remarks to the Author):

All my questions and comments have been addressed by the authors. The manuscript is now vastly improved with new data and additional discussion. I therefore recommend its publication in Nature Communications.

Reviewer #3 (Remarks to the Author):

The authors performed additional experiments to address my concerns. Although some attempts (growing FeSe on BaTiO₃) are not successful, I still appreciate the great efforts from the authors. The new experiment of surface doping (Rb doping on 2 unit-cell and 3 unit-cell FeSe layers) is beautiful, which demonstrates the competition between superconductivity and smectic phase. While the role of epitaxial strain on smectic phase in FeSe has yet to be clearly demonstrated in experiment, I think that the experimental discovery of this smectic phase in FeSe bi-layers and the experimental demonstration of the competition between superconductivity and smectic phase in FeSe thin films via carrier concentration provide sufficient new results for the publication in Nature Communications.

However, I do have an additional comment: as the author replied to my previous comment #4, the strain on FeSe from SrTiO₃ is not coherent, i.e. different from epitaxial strain (which has been widely studied in transition metal oxide thin films). That means that the strain in FeSe thin films is gradual and probably weaker compared to epitaxial strain which coherently fixes the in-plane lattice constant of the entire film. Then it implies that the role of strain on the smectic phase, if any, is delicate since the strain is weak and not uniform. I suggest that the authors add a new paragraph in the Discussion Section to explain that the strain effects on the smectic phase in FeSe has not been demonstrated in the current study, as well as why demonstrating this point is difficult in experiment. Considering that strain is relatively weak and gradual, its subtle role on the smectic phase in FeSe warrants further study in future. I hope this may stimulate other experimental groups (which have oxide-MBE) to clarify this important issue.

REVIEWERS' COMMENTS

Reviewer #1 (Remarks to the Author):

The authors have made a good effort to address my comments and included significantly more data to support their conclusions.

There is one aspect in the new section which I think needs clarification: In the new section, the authors discuss "good" and "bad" superconducting gap spectra - this should be qualified better. Is the selection of "good" and "bad" spectra done by eye, or by an algorithm? In either case, the criteria for a spectrum to be classed as "good" or "bad" need to be stated clearly, either in the main text or in the supplementary material - as otherwise it appears rather subjective. For example, for asymmetry of the gap, mentioned as a criterion, does this refer to the gap not having the same gap magnitude at positive and negative bias, the asymmetry of the coherence peaks (or gap edge), or both?

Once this point is addressed, I would support publication of the revised manuscript.

Reply: We thank the reviewer for thinking that we have made a good effort to improve our data and manuscript. In the revised manuscript, we have added a new section (Supplementary Note 11) to illustrate how the “good” and “bad” spectra are classified. The corresponding figures and statements are also presented below. We hope the reviewer’s concern can be fully addressed.

The good-superconducting group contains the dI/dV spectra which show energetically symmetric (with respect to zero energy) coherence peaks and absence of in-gap states, i.e. a fully opened U-shaped gap. Supplementary Figure 13a presents three examples for good superconducting gaps. FeSe thin films usually exhibit two pairs of energetically symmetric coherence peaks (as shown in the spectra #1 and #2 in Supplementary Fig. 13a). The spectra which only have one pair of symmetric coherence peaks are also classified in good superconducting group (#3 spectra in Supplementary Fig. 13a).

The bad superconducting group includes the spectra that show asymmetric superconducting gaps or contain in-gap states. If more than two pairs of peaks/features appear in the superconducting gap, they are recognized as in-gap states, and such spectra are sorted into bad superconducting group (see the three examples in Supplementary Fig. 13b).

In some cases, the coherence peaks and in-gap states are not symmetric in energies. For example, the spectrum #1 in Supplementary Fig. 13c shows a kink feature on the gap edge at negative energy; The spectrum #2 shows an in-gap state only at positive energy; The spectrum #3 show asymmetric coherence peaks at positive and negative energies. Such kind of spectra is also sorted into bad superconducting group.

Supplementary Figure 13 Examples for good and bad superconducting gaps. a Three typical dI/dV spectra in good superconducting group (set point, $V_s = 25$ mV, $I_t = 100$ pA). **b, c** Examples for bad superconducting gaps which exhibit in-gap states and asymmetric features, respectively (set point, $V_s = 25$ mV, $I_t = 100$ pA). The black arrows denote the coherence peaks and the red arrows denote the in-gap states/features.

Reviewer #2 (Remarks to the Author):

All my questions and comments have been addressed by the authors. The manuscript is now vastly improved with new data and additional discussion. I therefore recommend its publication in Nature Communications.

Reply: We appreciate that the reviewer thinks our manuscript is vastly improved and would like to recommend its publication.

Reviewer #3 (Remarks to the Author):

The authors performed additional experiments to address my concerns. Although some attempts (growing FeSe on BaTiO₃) are not successful, I still appreciate the great efforts from the authors. The new experiment of surface doping (Rb doping on 2 unit-cell and 3 unit-cell FeSe layers) is beautiful, which demonstrates the competition between superconductivity and smectic phase. While the role of epitaxial strain on smectic phase in FeSe has yet to be clearly demonstrated in experiment, I think that the experimental discovery of this smectic phase in FeSe bi-layers and the experimental demonstration of the competition between superconductivity and smectic phase in FeSe thin films via carrier concentration provide sufficient new results for the publication in Nature Communications.

Reply: We appreciate that the reviewer thinks our new result is beautiful and is sufficient to publish in Nature Communications.

However, I do have an additional comment: as the author replied to my previous comment #4, the strain on FeSe from SrTiO₃ is not coherent, i.e. different from epitaxial strain (which has been widely studied in transition metal oxide thin films). That means that the strain in FeSe thin films is gradual and probably weaker compared to epitaxial strain which coherently fixes the in-plane lattice constant of the entire film. Then it implies that the role of strain on the smectic phase, if any, is delicate since the strain is weak and not uniform. I suggest that the authors add a new paragraph in the Discussion Section to explain that the strain effects on the smectic phase in FeSe has not been demonstrated in the current study, as well as why demonstrating this point is difficult in experiment. Considering that strain is relatively weak and gradual, its subtle role on the smectic phase in FeSe warrants further study in future. I hope this may stimulate other experimental groups (which have oxide-MBE) to clarify this important issue.

Reply: The reviewer is right. We agree that the strain effects on the smectic phase of FeSe film has not been demonstrated in the current study. We hope that the future studies can clarify the strain effect on the formation of smectic phase. Accordingly, we add a new paragraph in the Discussion Section (the first paragraph of page 8) and also show it below.

“A promising way to decouple the charge and lattice degrees of freedom is to grow FeSe film on a substrate with even larger lattice constant. BaTiO₃ (BTO) could be a good candidate. Since the strength of tensile strain can be gradually tuned by the thickness of FeSe film, systematic study of FeSe/BTO may provide sufficient information to understand the role of strains for the formation of smectic phase. We are still working on the growth of FeSe films on BTO, however, it is challenging due to the large amount of defects on BTO substrate, which could be significantly improved by Oxide-MBE. The delicate effect of the strain warrants further theoretical and experimental investigations.”